# Anti-Influenza Activity of 6BIGOE: Improved Pharmacological Profile After Encapsulation in PLGA Nanoparticles

**DOI:** 10.3390/ijms26094235

**Published:** 2025-04-29

**Authors:** Josefine Schroeder, Jan Westhoff, Ivan Vilotijević, Oliver Werz, Stephanie Hoeppener, Bettina Löffler, Dagmar Fischer, Christina Ehrhardt

**Affiliations:** 1Section of Experimental Virology, Institute of Medical Microbiology, Center for Molecular Biomedicine (CMB), Jena University Hospital, Hans Knoell-Str. 2, 07745 Jena, Germany; josephine.schroeder@med.uni-jena.de; 2Division of Pharmaceutical Technology and Biopharmacy, Friedrich-Alexander-Universität Erlangen-Nürnberg, Cauerstr. 4, 91058 Erlangen, Germany; jan.westhoff@fau.de (J.W.); dagmar.fischer@fau.de (D.F.); 3Institute of Organic Chemistry and Macromolecular Chemistry, Friedrich Schiller University Jena, Humboldtstr. 10, 07743 Jena, Germany; ivan.vilotijevic@uni-jena.de (I.V.); s.hoeppener@uni-jena.de (S.H.); 4Department of Pharmaceutical and Medicinal Chemistry, Institute of Pharmacy, Friedrich Schiller University Jena, Philosophenweg 14, 07743 Jena, Germany; oliver.werz@uni-jena.de; 5Jena Center for Soft Matter (JCSM), Friedrich Schiller University Jena, Philosophenweg 7, 07743 Jena, Germany; bettina.loeffler@med.uni-jena.de; 6Institute of Medical Microbiology, Jena University Hospital, Am Klinikum 1, 07747 Jena, Germany; 7FAU NeW-Research Center for New Bioactive Compounds, Friedrich-Alexander-Universität Erlangen-Nürnberg, Nikolaus-Fiebiger-Str. 10, 91058 Erlangen, Germany

**Keywords:** influenza A virus (IAV), poly(D,L-lactic-co-glycolic acid) (PLGA), indirubin, nanoparticles (NPs), antiviral, glycogen synthase kinase 3 (GSK-3)

## Abstract

Influenza A virus (IAV) infections continue to threaten public health. Current strategies, such as vaccines and antiviral drugs, are limited due to their time-consuming development and drug-resistant strains. Therefore, new effective treatments are needed. Here, virus-supportive cellular factors are promising drug targets, and the encapsulation of candidate substances in poly(D,L-lactic-co-glycolic acid) (PLGA) nanoparticles (NPs) is intended to improve their bioavailability. This study investigates the potential of the indirubin derivative 6-bromoindirubin-3′-glycerol-oxime ether (6BIGOE), a glycogen synthase kinase 3 (GSK-3)β inhibitor, for its potential to regulate IAV replication in vitro. The effects of 6BIGOE-loaded PLGA NPs on cell metabolism were assessed by 3-(4,5-dimethylthiazol-2-yl)-2,5-diphenyltetrazolium bromide (MTT) and lactate dehydrogenase (LDH) assays in A549 and Calu-3 cells. Viral replication and spread were monitored in various IAV-infected cell lines in the absence and presence of free and 6BIGOE-loaded PLGA NPs via plaque assays and Western blot analysis. The encapsulation of 6BIGOE in PLGA NPs resulted in reduced negative side effects on cell viability while maintaining antiviral efficacy. Both encapsulated and free 6BIGOE exhibited antiviral activity, potentially through GSK-3β inhibition and the disruption of key signaling pathways required for viral replication. The data indicate 6BIGOE, particularly after encapsulation in NPs, as a potential candidate for further investigation and development as an antiviral agent to treat IAV infections.

## 1. Introduction

Seasonal influenza remains a major global health problem, with a high morbidity and mortality rate, responsible for an estimated annual burden of 3 to 5 million severe cases and 290,000 to 650,000 deaths [1,2]. In particular, elderly and immunocompromised patients are in danger of severe infection courses. Influenza A viruses (IAVs), which belong to the family of Orthomyxoviridae, are characterized by their single-stranded, negative-sensed RNA genome. Their high mutation rates, caused by the rapid genetic drift and shift in the virus, pose a challenge to control efforts, leading to seasonal outbreaks and occasional pandemics [3].

Although influenza vaccines are the most effective prophylactic measure, their low coverage and limited efficacy against newly emerging subtypes highlight the urgent need for efficient and broadly active antivirals to limit viral replication in infected hosts [1,4].

Currently, three classes of substances are approved by the Food and Drug Administration (FDA) and European Medicines Agency (EMA) for the therapeutic management of influenza virus infections: Matrix-2 (M2) ion channel inhibitors, neuraminidase inhibitors (NAIs) and the cap-dependent endonuclease inhibitor baloxavir marboxil [5,6]. The latter, as well as NAIs such as oseltamivir and peramivir, are active against both influenza A and B viruses, while M2 inhibitors, including amantadine and rimantadine, target only IAVs [6]. In any case, medication has to be applied in the early disease course to be effective. Additionally, the rapid emergence of resistance to these antiviral agents underscores the need for alternative therapeutic strategies with a high barrier to resistance [7].

Since influenza viruses hijack the host cell machinery for replication, targeting virus-supportive cellular factors offers promising intervention options. This approach has led to the development of host-directed therapies aiming at modulating these cellular factors rather than directly attacking the virus itself [8,9,10].

One such cellular factor gaining interest is glycogen synthase kinase 3 (GSK-3). GSK-3 is a serine/threonine kinase with two isoforms, GSK-3α and GSK-3β, which are expressed in most tissues and encoded by two genes, *GSK-3A* and *GSK-3B*. Although the two isoforms share high homology in their kinase domains, they differ at their C-termini [11,12]. Furthermore, the isoforms fulfill independent functions that cannot be compensated by another. The activation and deactivation of GSK-3 itself are controlled by the post-translational, site-specific phosphorylation of serine and tyrosine residues. GSK-3 regulation interferes with several key signaling pathways, including Wnt/β-catenin, nuclear factor-κB (NF-κB) and phosphatidylinositol 3-kinases (PI3Ks)/protein kinase B (Akt) with substrates such as the proline-rich Akt substrate of 40 kDa (PRAS40) and kinase with no lysine 1 (WNK1). The phosphorylation of serine 9 in GSK-3β or serine 21 in GSK-3α results in the inactivation of these kinases and the subsequent reversal of their inhibitory function. In contrast, the phosphorylation of tyrosine 216 in GSK-3β or tyrosine 279 in GSK-3α leads to their activation. Currently, around 100 proteins are believed to be GSK-3 substrates; however, they require initial phosphorylation by other kinases [11]. Initially identified for its role in regulating glycogen synthase in insulin signaling [13], GSK-3 has since been implicated in a variety of cellular processes, such as cell proliferation, differentiation, apoptosis and immune responses [14,15]. Therefore, the dysregulation of GSK-3-mediated signaling is implicated in several diseases, including neurodegenerative disorders like Alzheimer’s disease and metabolic disorders like type 2 diabetes, as well as in cancer progression. Concomitantly, GSK-3 inhibition shows therapeutic potential [16,17,18].

In context of viral infections, GSK-3 has been shown to play crucial but diverse roles. The literature reveals contradictory results, however, which reflect the complex interplay between GSK-3 and viral replication strategies [19,20]. In human immunodeficiency virus (HIV)-1 infection, GSK-3 expression is upregulated, potentially contributing to viral persistence [21]. In coxsackievirus B3 infections, increased GSK-3β activity triggers apoptosis and enhances viral release, while GSK-3β inhibition reduces these effects [22,23]. Furthermore, GSK-3 inhibition has been shown to suppress replication in varicella zoster virus [24] and coronaviruses by disrupting nucleocapsid phosphorylation [25]. During IAV infections, GSK-3 has also been shown to promote viral replication, especially by supporting viral entry [26].

When developing novel antiviral strategies, the use of existing pharmaceutical substances offers a significant advantage. The reuse of drugs with proven safety and established pharmacokinetics accelerates the further investigation process, reduces costs and minimizes the risk of side effects [27]. In this approach, natural products hold significant promise in drug repurposing approaches. An example is indirubin, a natural bis-indole alkaloid used in traditional Chinese medicine [28]. It has a broad spectrum of pharmacological effects, including anticancer, neuroprotective and anti-inflammatory effects, with potential applications in chronic inflammatory diseases, cancer, Alzheimer’s disease and metabolic disorders [28,29,30]. Interestingly, indirubin exhibits antiviral properties and inhibits viruses such as human adenoviruses [30], IAVs [31], HIV-1 [32], pseudorabies virus [33] and Japanese encephalitis virus [34].

Remarkably, the indirubin derivative 6-bromoindirubin-3′-glycerol-oxime ether (6BIGOE) is a potent modulator of inflammatory cytokines and lipid mediators in human monocytes by inhibiting GSK-3β, indicating potential therapeutic applications for inflammation-related diseases [35]. Therefore, it also could be effective during an ongoing influenza virus infection, limiting symptoms. Despite its efficacy, 6BIGOE exhibits high lipophilicity, poor water solubility, low bioavailability and potential cytotoxicity, limiting its therapeutic use [28,36]. Conventional administration methods are less efficient in achieving controlled release rates and targeting effects. To overcome these limitations, advanced drug delivery strategies such as crystal technology, surfactants and nanocarriers are used [37,38,39,40]. Among these, nanoparticle (NP)-based drug delivery systems have gained attention due to their ability to control drug release, increase drug bioavailability and reduce off-target effects [41]. In particular, poly(D,L-lactic-co-glycolic acid) (PLGA)-based NPs have been proven to be highly effective drug delivery vehicles due to their excellent biocompatibility, controlled release profiles and biodegradability [42,43]. PLGA NPs are an ideal choice to improve the bioavailability and therapeutic efficacy of hydrophobic drugs, ensuring that they are both stable and effective in clinical settings [44].

This study examined the effect of 6BIGOE on IAV replication in vitro. The data indicate that encapsulating the drug into PLGA NPs reduces its cytotoxicity while preserving its antiviral efficacy. The antiviral effect is presumably based on the 6BIGOE-mediated inhibition of GSK-3β activity that is necessary for efficient replication.

## 2. Results

### 2.1. Preparation and Characterization of PLGA NP [6BIGOE]

An emulsion–diffusion–evaporation (EDE) method was employed to encapsulate the small-molecule 6BIGOE (430.13 g mol^−1^) into PLGA NPs (Figure 1a). Dynamic light scattering revealed hydrodynamic diameters (HDs) of the NPs ranging from 185 to 194 nm, with polydispersity indices (PDIs) of 0.03 for PLGA NP [blank] and 0.12 for PLGA NP [6BIGOE], indicating monomodal particle size distributions. The zeta potential (ZP) of the PLGA NP [blank] was −22 mV, with a slight increase to approximately −14 mV observed after 6BIGOE encapsulation. The quantification of 6BIGOE via UV/Vis spectrophotometric measurement revealed a drug load (DL) of approximately 1.7%, resulting in an encapsulation efficiency (EE) of 34.8% (Figure 1b). The morphologies of both PLGA NP [blank] and PLGA NP [6BIGOE] were visualized using transmission electron microscopy (TEM) and revealed spherical and homogenous NPs (Figure 1c). In earlier in vitro studies, the release of 6BIGOE was investigated over a period up to 168 h in phosphate-buffered saline (PBS) supplemented with 1% Tween 80 (m/m) at pH 7.4, showing a fast and almost complete drug release already after 4 h [44,45].

### 2.2. PLGA NP [6BIGOE] Has Less Impact on Cell Metabolic Activity than Free 6BIGOE

To identify the non-toxic concentrations of encapsulated and free 6BIGOE suitable for the in vitro treatment of A549 and Calu-3 cells, 3-(4,5-dimethylthiazol-2-yl)-2,5-diphenyltetrazolium bromide (MTT) assays were performed (Figure 2a,b). Only viable cells reduce the tetrazolium dye MTT to the insoluble formazan, measurable at 562 nm. For this purpose, the cells were treated with the drugs at the indicated concentrations for 24 h in comparison to solvent controls, which were arbitrarily set to 100%. In the case of 6BIGOE, dimethyl sulfoxide (DMSO) served as a control, while H_2_O was used for the PLGA NPs. Lactate dehydrogenase (LDH) release measurements complemented these assays to validate the results, as LDH is released by dying cells devoid of cell membrane integrity (Figure 2c,d). In the A549 cells, exposure to 2 µM of free 6BIGOE significantly reduced metabolic activity and increased LDH release to 30% (Figure 2a,c). Interestingly, treatment with PLGA NP [6BIGOE] affected neither metabolic activity nor LDH release, even at concentrations up to 4 µM. In the Calu-3 cells, treatment with all tested concentrations of both the encapsulated and free 6BIGOE (0.03–4 µM) had no impact on metabolic activity (Figure 2b). However, exposure to 4 µM of free 6BIGOE resulted in an 11% increase in LDH release (Figure 2d), whereas PLGA NP [6BIGOE] showed no effect on cytotoxicity at any tested concentration. Notably, PLGA NP [blank] had no impact on A549 and Calu-3 cell viability in any of the experiments (Appendix A).

Since 6BIGOE is known to affect GSK-3β phosphorylation, phospho-GSK-3β (pGSK-3β) levels were monitored by Western blot analysis and the impacts of free and encapsulated 6BIGOE were analyzed. Pictilisib, a known inhibitor of the PI3K signaling pathway, which plays a crucial role in regulating GSK-3 activity, served as a positive control. The treatment of A549 (Figure 2e) and Calu-3 (Figure 2f) cells with either free or PLGA-encapsulated 6BIGOE resulted in significantly reduced pGSK-3β levels within each concentration step, from 0.05 to 0.5 μM for A549 cells and from 0.1 to 1 µM for Calu-3 cells.

The results indicate that encapsulated 6BIGOE has lower effects on cellular metabolic activity and cell survival compared to free 6BIGOE, while the 6BIGOE-mediated effects on GSK-3β phosphorylation are comparable. Overall, these data suggest that A549 cells are more sensitive to treatment with free 6BIGOE than Calu-3 cells, which can tolerate slightly higher concentrations. Consequently, concentrations of up to 1 µM of both free and encapsulated 6BIGOE were chosen for further experiments in epithelial cells.

### 2.3. 6BIGOE Treatment of IAV-Infected Cells Results in Reduced Viral Titers In Vitro and Is Associated with Inhibition of GSK-3β-Mediated Signaling Processes

To further investigate the effects of 6BIGOE on IAV replication, multi-cycle viral growth experiments were performed in A549 and Calu-3 cells using three different IAV strains: H1N1 strain A/Puerto Rico/8/34 (PR8) and the H1N1pdm isolates A/NRW/173/09 (NRW173) and A/Jena/5258/09 (Jena5258). After an initial infection of 30 min and treatment with 0.1 and 1 µM of both free and encapsulated 6BIGOE in A549 (Figure 3a,c,e) and Calu-3 cells (Figure 3b,d,f) for 24 h, progeny infectious virus particles were quantified by plaque assay. Independent of the cell line used, treatment with both 6BIGOE and PLGA NP [6BIGOE] resulted in a concentration-dependent reduction in virus titers for all three virus strains, with the inhibitory effect being slightly more pronounced for free 6BIGOE. Thus, these data indicate that 6BIGOE inhibits the replication of IAV.

To confirm the inhibitory effect of 6BIGOE on viral replication and to further examine possible cellular targets, infection experiments were carried out within the first replication cycle of IAV. Here, the expressions of IAV hemagglutinin (HA) and polymerase basic protein 1 (PB1) were monitored at 4 h, 6 h and 8 h post infection (p.i.) by Western blot analysis in the A549 (Figure 3g) and Calu-3 cells (Figure 3h), and protein expression was quantified (Appendix A). Furthermore, the effects of IAV infection on GSK-3β phosphorylation and 6BIGOE-mediated effects were analyzed compared to the solvent control (DMSO). In the A549 cells, the expression of HA was already detectable at 6 h p.i. and was reduced in presence of 6BIGOE. However, this reduction was no longer apparent at 8 h p.i., while PB1 levels remained suppressed at this time point. In Calu-3 cells, viral protein expression seemed to be delayed. Nonetheless, viral protein expression was detectable at 8 h p.i., with a visible reduction in PB1 and HA levels following 6BIGOE treatment in comparison to the DMSO-treated control. In the presence of 6BIGOE, the IAV-induced GSK-3β phosphorylation, particularly at 8 h p.i., was visible, correlating to the reduction in viral protein expression. During IAV infection, WNK1 and PRAS40 phosphorylation occurred in both cell lines, notably at 8 h p.i., and this activation was reduced by treatment with 6BIGOE.

These findings indicate that treatment of IAV-infected epithelial cells with 6BIGOE and PLGS NP [6BIGOE] at non-toxic concentrations effectively reduces viral load and propagation by the regulation of GSK-3β-mediated signaling.

## 3. Discussion

Given the global health threat associated with IAV infections, effective treatment strategies are essential to mitigate the significant number of infections and associated deaths. Targeting host intracellular factors that support virus replication is of particular interest, as this reduces the risk of resistance development. Furthermore, the repurposing of existing therapeutics and substances with pharmaceutical effects saves both development costs and time. This study examines the potential use of 6BIGOE as a GSK-3β inhibitor against IAV infections.

However, due to the physicochemical properties of bioactive small molecules such as indirubin derivatives, several associated challenges emerge, including poor water solubility [46,47] and membrane permeability [28]. Additionally, the drug’s short half-life in blood of nearly 1 h after oral administration in mice [48] and cytotoxic effects against different cell lines [36] are further disadvantages. These issues can be addressed by encapsulating the drug in polymeric NPs made of PLGA, which is approved by the FDA and EMA for different controlled-drug-release systems. A similar approach was employed by Czapka et al., where encapsulating 6BIGOE repressed the off-target effects of the free drug on human monocytes [44].

In order to overcome such drug-related challenges, 6BIGOE was encapsulated into spherical PLGA NPs with HDs below 200 nm and narrow and homogeneous particle size distributions. The HD, PDI, ZP and EE of the formulated PLGA NPs were found to be comparable to those reported in previous studies [44,45]. The increase in ZP after the encapsulation of 6BIGOE may be attributed to a part of the drug being located on or near the particle surface, since the drug is likely uncharged at a neutral pH. This observation correlates with the burst release observed in drug release studies previously conducted for PLGA NP [6BIGOE] [44].

6BIGOE exhibited a greater impact on A549 cells than on Calu-3 cells, which could tolerate up to 2 µM of free 6BIGOE. While both cell lines are human lung adenocarcinoma cells, they differ significantly in characteristics [49]. Calu-3 cells form a thin mucus layer resembling the bronchial submucosal glands [50], whereas A549 cells, lacking this feature, represent type II alveolar cells and may be more sensitive to external agents [51]. Our findings indicate that the encapsulation of 6BIGOE in PLGA NPs reduced the metabolic effects and LDH release in both lung epithelial cell lines. This may be due to the slower release of 6BIGOE from the NPs, resulting in lower cytotoxicity. The incorporation of 6BIGOE into PLGA NPs might lower their effective local concentration; but intracellular concentrations appeared comparable, as indicated by similar effects observed on GSK-3β phosphorylation. Additionally, off-target interactions with outer membranes, linked to cytotoxicity and premature drug degradation, could potentially be reduced. PLGA NP [6BIGOE] demonstrates potential for antiviral drug delivery in vitro, but its in vivo efficacy remains underexplored. Given the highly lipophilic nature of 6BIGOE, its direct administration in humans is not feasible; however, encapsulation within PLGA NPs makes its application in vivo achievable. One study reported significantly enhanced circulation times for indirubin derivatives, resulting in a 984% increase in bioavailability compared to their free form, following oral administration in rats [52]. While surface modifications enhance targeted delivery to infection sites [53], challenges such as optimal drug concentration, immune response and pharmacokinetics require further investigation.

PLGA NPs are already known to deliver poorly soluble drugs into tissues like tumors [54,55] or epithelial cells [56], enhancing their effect. In this study, the antiviral effect of 6BIGOE was slightly higher compared to PLGA NP [6BIGOE]. This difference could potentially be explained by the delayed release caused by the diffusion barrier introduced with the addition of PLGA NPs. Nevertheless, the cytotoxic effects of 6BIGOE were attenuated by the encapsulation, suggesting that higher doses of encapsulated 6BIGOE could be administered, even for a longer period of time.

In the present study, PLGA NP [6BIGOE] was used as a proof-of-concept for antiviral therapy, with a focus on biological experiments. While administration aspects require further investigation, both inhalation and injection routes remain viable options for this type of polymer NP. Previous studies have investigated the pulmonary delivery of PLGA-based NPs via inhalation for the treatment of cystic fibrosis [57,58], asthma [59] and lung cancer [60]. Systemic administration has also been explored for enhancing the delivery and efficacy of therapeutic agents across various medical applications, providing prolonged circulation, targeted delivery, improved therapeutic outcomes and maintaining safety and biocompatibility [44].

Indirubins are recognized for their antiviral properties against various viruses, including adenoviruses [30] and IAVs [31]. Derivatives such as indirubin-3′-(2,3-dihydroxypropyl)-oxime ether and indirubin-3′-oxime (I3O) have already been tested in terms of their anti-inflammatory and antiviral effects in endothelial cells and macrophages [31,61]. Infection with a H9N2 or H5N1 IAV strains increased cytokine and chemokine release, which was suppressed in the presence of indirubin derivatives. In these studies, the roles of mitogen-activated protein kinases (MAPKs) and activator of transcription 3 (STAT3) proteins in reducing indirubin-mediated anti-inflammatory effects were discussed [61]. Although this study focused exclusively on H1N1 strains to evaluate the efficacy of 6BIGOE, the promising effects observed for indirubin derivatives with H9N2 and H5N1 strains in other studies [31,61] indicate the potential usability of 6BIGOE against other influenza virus strains. However, this would need to be further investigated in the future. Interestingly, 6BIGOE has already been shown to counteract GSK-3β during inflammation in human monocytes by reducing the TLR4-induced release of pro-inflammatory mediators, including IL-1β, IL-6, TNFα, IL-8 and PGE_2_ [35]. While the present study did not assess the anti-inflammatory properties of 6BIGOE, further research is warranted to investigate its potential to counteract IAV-induced cytokine storms in vitro, particularly in various immune cells and lung epithelial cell lines. I3O has been shown to delay viral replication by inhibiting viral gene transcription and translation [31]. In our study, treatment with 6BIGOE resulted in the delayed expression of HA protein in A549 cells at 6 h p.i., but at 8 h p.i., the viral protein expression had caught up and was similar to that in the DMSO control.

GSK-3β is a regulatory factor of various signaling pathways, including PI3K/Akt-mediated signaling, and GSK-3β is induced during viral replication [20,62,63]. Since PI3K/Akt signaling has supportive functions during IAV replication, it is not surprising that the inhibition of GSK-3β with 6BIGOE also results in reduced viral load. The use of non-toxic concentrations of 6BIGOE suggests that it selectively inhibits viral replication rather than disrupting cellular processes. RNA interference screening identified 295 cellular host factors that play a crucial role in influenza virus replication, including GSK-3β, though its exact mechanistic role remains unclear [10]. Previous studies reported that GSK-3 phosphorylates a key residue on the GTPase dynamin I which is involved in clathrin-mediated endocytosis [64,65]. By inhibiting GSK-3β, the formation and maturation of clathrin-coated pits are promoted [64,66]. Considering the importance of receptor-mediated endocytosis in IAV entry, further research is necessary to evaluate whether inhibiting GSK-3β with 6BIGOE could disrupt endocytic processes and ultimately limit viral entry into host cells. This may occur because the enhanced formation of clathrin-coated pits could accelerate the internalization of surface receptors, leaving fewer available for virus binding and entry.

Overall, the PI3K/Akt signaling pathway, including GSK-3β, is a promising target for influenza virus intervention [63,67,68], as its inhibition interferes with replication processes, from viral entry to release. In this study, IAV infection also activated PRAS40 and WNK1, although their precise role in IAV replication is largely unknown. PRAS40 has a regulatory function at the intersection of the Akt/mammalian target of rapamycin (mTOR) signaling pathway [69,70,71]. mTOR is a highly conserved serine/threonine kinase that is part of two distinct multi-protein complexes, mTOR complex 1 (mTORC1) and 2 (mTORC2), which are well studied in diseases such as cancer and type 2 diabetes [72,73,74]. mTORC1 itself plays a pivotal role in the regulation of cellular protein synthesis, proliferation and metabolism, particularly in response to nutrients and growth factors [72]. Interestingly, mTORC1 activation has recently been implicated in facilitating influenza virus replication, highlighting its potential role as a host factor in viral infections. However, the role of mTORC2 in IAV infection remains poorly understood and warrants further investigation [75,76]. The relationship between the PI3K/Akt cascade and the WNK protein kinase signaling pathway, which primarily regulates ion transport across cell membranes, has been extensively studied in the context of cancer [77,78,79]. PI3K/Akt signaling can activate the WNK-oxidative stress responsive kinase 1 (OSR1)/PS/Ste20-related proline–alanine-rich kinase (SPAK)-NaCl cotransporter (NCC) pathway in a hyperinsulinemic db/db mouse model, providing critical evidence of the Akt-mediated kinase activation of WNK [80]. Although the regulation and function of PRAS40 or WNK1 during IAV infection has not yet been elucidated, the interaction with the PI3K/Akt signaling pathway might explain the reduced phosphorylation of PRAS40 and WNK1 through 6BIGOE-mediated GSK-3β inhibition.

## 4. Materials and Methods

### 4.1. Cell Culture, Viruses and Substances

A human epithelial lung cancer cell line (A549) was cultured in Dulbecco’s Modified Eagle’s Medium (DMEM; Anprotec, Bruckberg, Germany) with high glucose, stable glutamine and sodium pyruvate supplemented with 10% fetal calf serum (FCS; Anprotec, Bruckberg, Germany) at 37 °C and 5% CO_2_. Madin–Darby canine kidney II (MDCK II) and human lung adenocarcinoma cell line (Calu-3) cells were grown in Minimum Essential Medium with Earle’s salts and L-glutamine (EMEM; Anprotec, Bruckberg, Germany) with 10% FCS. A549 and Calu-3 cells were utilized in this study, as they closely resemble lung cells in culture, making them an excellent in vitro model for infection experiments and drug testing due to their high susceptibility to infection. The influenza virus strains used in this study were A/Puerto Rico/8/34 (PR8; H1N1), A/NRW/173/09 (NRW173; H1N1pdm) and A/Jena/5258/09 (Jena5258; H1N1pdm). 6BIGOE (Prof. Dr. Ivan Vilotijević, Friedrich Schiller University Jena, Jena, Germany) and pictilisib (Selleckchem, Cologne, Germany) were dissolved in DMSO (Sigma-Aldrich, Darmstadt, Germany) for biological experiments.

### 4.2. Preparation of PLGA NPs

6BIGOE was synthesized following a previously reported procedure [45]. The characterization data for the synthetic 6BIGOE matched the previously reported data, and purity was confirmed by elemental analysis. PLGA NPs were formulated using an EDE technique. 25 mg PLGA (Resomer^®^ RG 502; M_W_ 7000–17,000 g mol^−1^, ester terminated, 50:50 lactide/glycolide ratio; Sigma-Aldrich, Darmstadt, Germany) and 1.25 mg 6BIGOE were dissolved in 2.5 mL dichloromethane (≥99.8%; Thermo Scientific™, Waltham, MA, USA) as the organic phase. For the formation of an oil-in-water (O/W) emulsion, the organic phase was added dropwise at room temperature (RT) under magnetic stirring (400 rpm) to 7.5 mL of an aqueous phase consisting of 2% (*w*/*v*) poly(vinyl alcohol) (30,000–70,000 g mol^−1^; Sigma-Aldrich, Darmstadt, Germany) as stabilizer in deionized H_2_O at pH 7.4. The emulsion was sonicated (Branson Sonifier Cell Disruptor B15, Branson Ultrasonic Corp., Danbury, CT, USA) for 90 s. Afterwards, the resulting nanoemulsion was diluted with deionized H_2_O to facilitate the evaporation of dichloromethane under stirring at RT overnight. The NP dispersion was washed thrice with deionized H_2_O by centrifugation (22,000× *g*, 20 min; Beckmann Coulter Allegra 64R, Brea, CA, USA).

### 4.3. Laser Light Scattering Techniques

A Zetasizer Ultra (Malvern Instruments Ltd., Malvern, UK) was used for NP characterization regarding HD, PDI and ZP in H_2_O at 25 °C (refractive index 1.33, viscosity 0.887 mPa × s). The measurement of HD and PDI were conducted using dynamic light scattering (173°, five runs per sample) in ZEN0040 cuvettes (Brand, Wertheim, Germany). ZP was examined as electrophoretic mobility in deionized H_2_O with three runs per sample in a DTS 1070 folded capillary cell (Malvern Instruments). The data of three independent batches were analyzed using the Malvern Instruments Zetasizer software ZS Xplorer v1.3.1.7 and expressed as mean ± standard deviation (SD).

### 4.4. Transmission Electron Microscopy

For TEM, samples were stained with 2% phosphotungstic acid solution (Thermo Scientific™, Waltham, MA, USA) on formvar carbon-coated copper grids (Electron Microscopy Sciences, Hatfield, PA, USA). Images were acquired on a Tecnai G^2^ 20 (FEI, Eindhoven, The Netherlands) at an acceleration voltage of 120 kV. Images were recorded on a MegaView (OSIS, Olympus Soft Imaging Systems, Tokio, Japan) CCD camera with 1376 × 1024 image pixels. Images were optimized for contrast utilizing Fiji ImageJ 1.54i [81].

### 4.5. Quantification of 6BIGOE by UV/Vis Spectrophotometric Measurement

The amount of encapsulated 6BIGOE was determined by UV/Vis spectrophotometric measurement according to the method of Czapka et al. [35]. Lyophilized NPs (72 h, −20 °C, 0.1 mbar; 25 L Genesis SQ EL-85, SP Scientific, Warminster, PA, USA) were dissolved in DMSO (≥99.8%; Carl Roth, Karlsruhe, Germany) and analyzed by a Tecan Spark 10M plate reader (Tecan Group, Männedorf, Switzerland) at 520 nm. Calibration was performed with linear dilutions of 6BIGOE in DMSO between 10 and 150 µg mL^−1^ (r^2^ = 0.999). The DL [%] was calculated by the ratio of encapsulated drug (mg) per 100 mg of NPs by weighting. The ratio of the determined DL to the theoretical DL corresponds to the EE [%]. The SparkControl v3.2 software (Tecan Group, Männedorf, Switzerland) was used for measurements, and results were presented as mean ± SD from three independent experiments.

### 4.6. Assessing Drug Impact on Cellular Metabolic Activity and Cytotoxicity (MTT and LDH Assays)

To determine the impact of free and encapsulated 6BIGOE (PLGA NP [6BIGOE]) on cellular metabolic activity, MTT assays were performed and LDH release was measured to examine cytotoxic effects. A549 (10,000 cells/well) or Calu-3 cells (100,000 cells/well) were seeded into 96-well plates 24 h prior to use. The cells were treated with the indicated concentrations of 6BIGOE and PLGA NP [6BIGOE] or equivalent volumes of PLGA NP [blank] and the solvent controls (DMSO for 6BIGOE and H_2_O for PLGA NPs) in 100 µL of DMEM (A549 cells) or EMEM (Calu-3 cells) supplemented with 10% FCS. Cells were incubated at 37 °C with 5% CO_2_. To ensure accurate comparisons, the volume of PLGA NP [blank] was adjusted to match the suspension volumes administered to the PLGA NP [6BIGOE]-treated cells. For PLGA NP [blank], the quantity was presented as a percentage, representing the volume of PLGA NP [blank] relative to the medium. After 24 h, supernatants were collected for LDH assay and stored at −20 °C until further use. For the maximum LDH release control, 10 µL of lysis buffer provided by the CyQUANT^TM^ LDH Cytotoxicity Assay kit (Thermo Scientific™, Waltham, MA, USA) was added for 45 min at 37 °C and 5% CO_2_ after 24 h of untreated growth. For the MTT assay 100 µL of EMEM (Calu-3) or DMEM (A549) containing MTT (Sigma-Aldrich, Darmstadt, Germany) at a final concentration of 1 mg mL^−1^ was added to the cells, followed by 2 h of incubation. The medium was then removed, and cells were lysed in 30 µL of DMSO. Absorbance was measured at 562 nm using a FLUOstar Omega plate reader (BMG Labtech, Ortenberg, Germany) with the corresponding Omega series software v5.50 R4. LDH release was measured from the supernatants by using the CyQUANT^TM^ LDH Cytotoxicity Assay kit according to the manufacturer’s protocol. LDH release from compound-treated cells was compared to maximum LDH release from lysis buffer-treated control cells, which were set to 100%, and medium-treated samples were used as spontaneous LDH release controls and represented 0%. Absorbance was measured at 490 nm and 680 nm using a FLUOstar Omega plate reader (BMG Labtech, Ortenberg, Germany) with the corresponding Omega series software v5.50 R4. The background absorbance at 680 nm was subtracted from that at 490 nm.

### 4.7. Viral Infection

A549 cells were seeded in 12-well plates (250,000 cells/well) 24 h prior to viral infection. Calu-3 cells were seeded in 12-well plates (500,000 cells/well) 36 h prior to viral infection. The cells were washed with PBS (Carl Roth, Karlsruhe, Germany) prior to infection and either left uninfected or infected with the indicated multiplicity of infection (MOI) of IAV in 250 µL of PBS/BA (PBS supplemented with 0.2% BSA (Carl Roth, Karlsruhe, Germany), 1 mM MgCl_2_ (Sigma-Aldrich, Darmstadt, Germany) and 0.9 mM CaCl_2_ (Sigma-Aldrich, Darmstadt, Germany)) for 30 min at 37 °C and 5% CO_2_.

Subsequently, the cells were incubated in DMEM/BA (A549 cells) or MEM/BA (Calu-3 cells) containing the indicated substances until the designated time points p.i. DMEM/BA and MEM/BA consisting of DMEM or MEM, respectively, were supplemented with 0.2% BSA, 1 mM MgCl_2_, 0.9 mM CaCl_2_ and 0.2 µg ml^−1^ TPCK trypsin (Sigma-Aldrich, Darmstadt, Germany).

### 4.8. Plaque Assay

Viral titers were determined by plaque assay. MDCK II cells (2,000,000 cells/well) were seeded in 6-well plates 24 h before infection to achieve a confluent layer. The cells were washed with PBS and infected with serial dilutions of the respective samples in PBS/BA supplemented with 100 U mL^−1^/0.1 mg mL^−1^ penicillin/streptomycin (pen/strep; Anprotec, Bruckberg, Germany) for 30 min at 37 °C and 5% CO_2_. Afterwards, virus solutions were discarded and replaced by 2 mL prewarmed soft agar containing MEM (Gibco™, Waltham, MA, USA) supplemented with 0.2% bovine serum albumin (BSA), 0.01% DEAE Dextran (Pharmacia Biotech, Berlin, Germany), 0.2% NaHCO_3_ (Gibco™, Waltham, MA, USA), 100 U mL^−1^/0.1 mg mL^−1^ pen/strep, 0.2 µg mL^−1^ TPCK trypsin and 0.9% agar (Oxoid, Wesel, Germany). The plates were incubated at 37 °C and 5% CO_2_ for three days. Plaque forming units (PFUs) were visualized using neutral red (Sigma-Aldrich, Darmstadt, Germany) staining and counted.

### 4.9. Western Blot Analysis

To analyze protein expression, Western blot analysis was performed. Triton lysis buffer (20 mM Tris-HCl (pH 7.4), 137 mM NaCl, 10% glycerol, 1% Triton-X-100, 2 mM EDTA, 50 mM β-glycerophosphate, 20 mM sodium pyrophosphate) containing protease inhibitors (0.2 mM Pefabloc^®^ (Sigma-Aldrich, Darmstadt, Germany), 5 µg mL^−1^ aprotinin (Carl Roth, Karlsruhe, Germany), 5 µg mL^−1^ leupeptin (Sigma-Aldrich, Darmstadt, Germany), 1 mM sodium vanadate (Sigma-Aldrich, Darmstadt, Germany) and 5 mM benzamidine (Sigma-Aldrich, Darmstadt, Germany) was used to lyse the cells for 30 min at 4 °C. Cell lysates were cleared via centrifugation (14,000 rpm, 10 min, 4 °C) and protein content was determined using Protein Assay Dye Reagent Concentrate (BioRad, Benicia, CA, USA). Lysates and protein marker (PageRuler™ Prestained Protein Ladder, Thermo Scientific™, Waltham, MA, USA) were loaded on a 7% or 10% SDS-PAGE and afterwards blotted onto a 0.2 µm nitrocellulose membrane (Amersham™ Protran^®^, Marlborough, MA, USA). Membranes were blocked for 1 h with 2% milk. Proteins were detected using the primary antibodies (1:1000) listed in Table 1. The secondary antibodies used were WesternSure^®^ goat anti-rabbit HRP and WesternSure^®^ goat anti-mouse HRP (LI-COR^®^ Bioscience, Bad Homburg, Germany) at dilutions of 1:5000. Membranes were incubated in Pierce^®^ ECL Western Blotting Substrate (Thermo Scientific™, Waltham, MA, USA) and further developed with the Fusion©FX6.Edge (Vilber Lourmat, Eberhardzell, Germany). Protein levels were quantified and normalized to the α-tubulin or heat shock protein 90 (HSP90) loading controls using Fiji ImageJ 1.54i [81].

## 5. Conclusions

In conclusion, this study highlights 6BIGOE as a promising candidate for anti-influenza intervention. 6BIGOE encapsulated into PLGA NPs exhibits lower cytotoxicity and impact on cellular metabolic activity compared to its free form, while maintaining effective GSK-3β inhibition. 6BIGOE and PLGA NP [6BIGOE] show antiviral properties against H1N1 IAV strains in lung epithelial cells, likely through the modulation of GSK-3β signaling. To further elucidate the mechanisms mediated by GSK-3β during IAV infection, additional research is required. Moreover, investigations into more complex technical systems are essential.

## Figures and Tables

**Figure 1 ijms-26-04235-f001:**
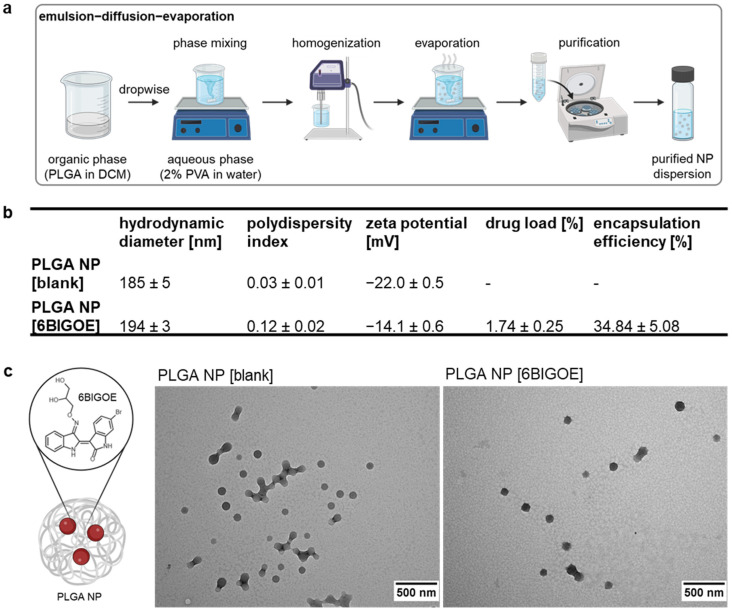
6BIGOE was successfully encapsulated into PLGA NPs. (**a**) PLGA NPs were prepared by an emulsion–diffusion–evaporation (EDE) method. (**b**) The table shows the particle size (hydrodynamic diameter (HD)), polydispersity index (PDI), zeta potential (ZP), drug load (DL) and encapsulation efficiency (EE) of PLGA NP [blank] and PLGA NP [6BIGOE]. (**c**) The chemical structure of 6BIGOE encapsulated into a PLGA NP. Transmission electron microscopy (TEM) shows the PLGA NP [blank] and PLGA NP [6BIGOE] samples. The scale bars represent 500 nm. The data are means ± standard deviation (SD) (**b**) or representations (**c**) of three independent batches.

**Figure 2 ijms-26-04235-f002:**
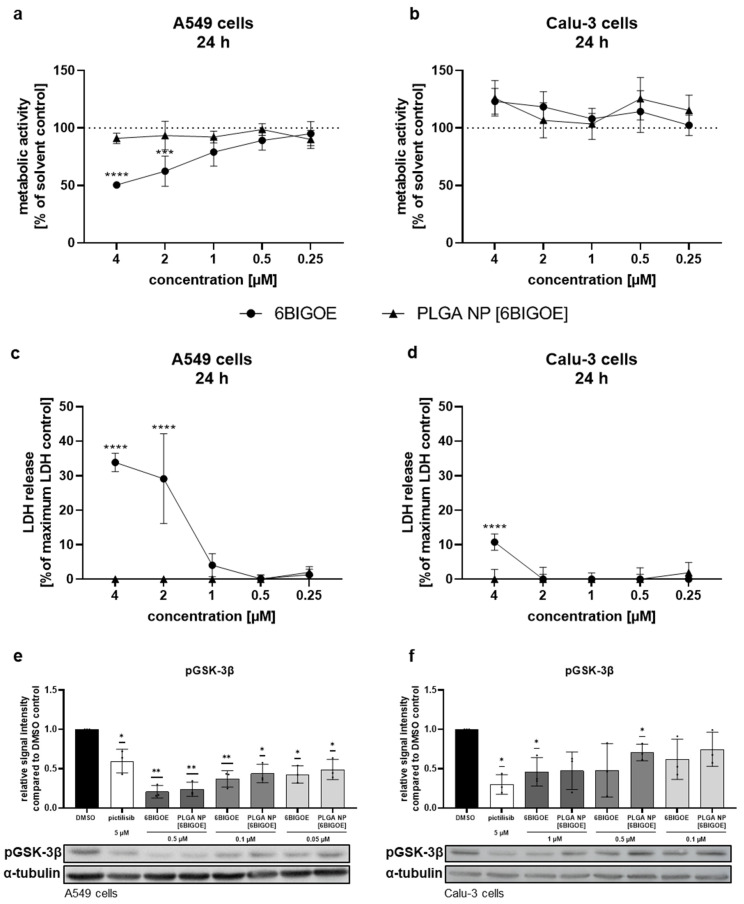
PLGA NP [6BIGOE] had less impact on cellular metabolic activity than free 6BIGOE. A549 (**a**,**c**,**e**) and Calu-3 (**b**,**d**,**f**) cells were treated with the indicated concentrations of 6BIGOE or PLGA NP [6BIGOE] for 24 h (**a**–**d**) or 18 h (**e**,**f**). (**a**,**b**) The metabolic activity was measured using MTT assays and normalized to the solvent-control-treated samples (DMSO, H_2_O). The data are the means ± SD of four independent experiments with two technical replicates. The values are given as percentages of the solvent-treated control results. Statistical significance was determined by two-way ANOVA with Dunnett’s multiple comparisons test. *** *p* < 0.001; **** *p* < 0.0001. (**c**,**d**) LDH levels were measured by using cell supernatants after 24 h of treatment with the indicated substances. The lysis-buffer-treated sample served as the maximum LDH release control and was arbitrarily set to 100%; medium-treated samples were used as spontaneous LDH release controls and represent 0%. The data are the means ± SD of three independent experiments with two technical replicates. The values are given as percentages of the maximum LDH release control. Statistical significance was determined by two-way ANOVA with Dunnett’s multiple comparisons test. **** *p* < 0.0001. (**e**,**f**) The phosphorylation status of GSK-3β was measured with Western blot analysis in A549 (**e**) and Calu-3 (**f**) cells for DMSO, pictilisib and different concentrations of 6BIGOE and PLGA NP [6BIGOE]. Quantification was performed by using α-tubulin as a loading control via Fiji ImageJ 1.54i. The data are the means ± SD of three independent experiments. Statistical significance was determined by a one-sample *t*-test comparing to a hypothetical mean of 1. * *p* < 0.05; ** *p* < 0.01.

**Figure 3 ijms-26-04235-f003:**
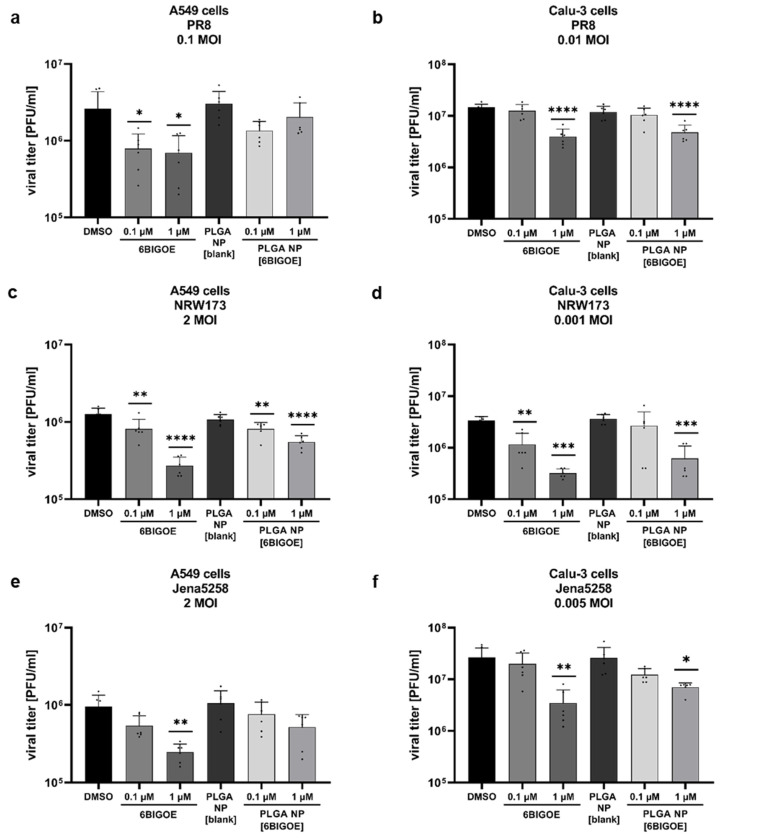
PLGA NP [6BIGOE] treatment causes an inhibition of IAV replication in vitro through the inhibition of GSK-3β. A549 (**a**,**c**,**e**,**g**) and Calu-3 (**b**,**d**,**f**,**h**) cells were infected with the indicated IAV strains (PR8 (**a**,**b**,**g**,**h**), NRW173 (**c**,**d**) or Jena5258 (**e**,**f**)) for 30 min and were subsequently treated with 0.1 and 1 µM (**a**–**f**), 0.5 µM (**g**) or 1 µM (**h**) of 6BIGOE, PLGA NP [6BIGOE] or controls (DMSO, PLGA NP [blank]). (**a**–**f**) After 24 h of incubation, supernatants were collected and progeny virus titers (plaque forming units (PFUs)/mL) were determined by plaque assay. The means + SD of three independent experiments with two technical replicates are depicted. Statistical significance was determined using one-way ANOVA and Dunnett’s multiple comparisons test. * *p* < 0.05; ** *p* < 0.01; *** *p* < 0.001; **** *p* < 0.0001. (**g**,**h**) Cell lysates were harvested at 4 h, 6 h and 8 h p.i. for Western blot analysis. The effects of 6BIGOE on viral HA and PB1, as well as on the phosphorylation of PRAS40, WNK1 and GSK-3β, are presented. A representative example and the densitometry (n-fold over 8 h DMSO infected for HA, PB1 or n-fold over 4 h DMSO uninfected for pPRAS40, pWNK1 and pGSK-3β) of three independent experiments are depicted. Statistical significance was assessed using an unpaired *t*-test to compare 6BIGOE-treated samples (+) with solvent-treated (−) samples at each time point. * *p* < 0.05; ** *p* < 0.01; **** *p* < 0.0001.

**Table 1 ijms-26-04235-t001:** Primary antibodies for Western blot analysis.

Target	Manufacturer
IAV (HA)	GeneTex (GTX127357)
IAV (PB1)	GeneTex (GTX125923)
α-tubulin	Cell Signaling Technology (2125)
HSP90	Cell Signaling Technology (4877)
pGSK-3β (Ser9)	Cell Signaling Technology (5558)
pPRAS40 (Thr246)	Cell Signaling Technology (13175)
pWNK1 (Thr60)	Cell Signaling Technology (4946)

## Data Availability

The data are contained within the article and Appendix A. The original Western blot pictures were submitted with this article.

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
