# Peer review of "Anti-Influenza Activity of 6BIGOE: Improved Pharmacological Profile After Encapsulation in PLGA Nanoparticles"

_ijms, 2025, doi:10.3390/ijms26094235_

Round 1

Reviewer 1 Report

Comments and Suggestions for Authors

This paper evaluated the antiviral potential of the GSK-3β inhibitor 6BIGOE against influenza A virus (IAV), both in free form and encapsulated in PLGA nanoparticles (NPs), to improve bioavailability and reduce cytotoxicity. This study highlights PLGA-encapsulated 6BIGOE as a promising host-directed antiviral candidate, combining improved safety and efficacy for IAV therapy. The work is well-structured, with clear experimental validation, and addresses an important gap in IAV therapeutics. However, some methodological, mechanistic and discussion issues need to be clarified to strengthen the conclusions.

Minor Comments:

  1. In Figure 2, does the concentration of PLGA NP [6BIGOE] refer to the concentration of encapsulated 6BIGOE in the nanoparticles? If so, how did the authors quantify this concentration? The Materials and Methods section should provide a detailed description of the quantification method for 6BIGOE concentration in PLGA NP [6BIGOE]. This is critical for comparing the effects of encapsulated versus free 6BIGOE on metabolic activity. Only if the concentrations of 6BIGOE are consistent can the beneficial effects of encapsulation be accurately evaluated. If the concentration here refers to the overall concentration of PLGA NP [6BIGOE], then the comparison between 6BIGOE and PLGA NP [6BIGOE] in Figures 2 and 3 would be meaningless.
  2. In Figures 3a-f, the antiviral effect of PLGA NP [6BIGOE] appears weaker than that of free 6BIGOE. What is the primary reason for this observation? This result seems contradictory to the statement in the Discussion: "In addition, encapsulating small molecules into NPs may have a more significant impact in vivo by enhancing circulation time. For instance, an encapsulated indirubin derivative was 984% more bioavailable than its free form following oral administration in rats." The authors should address and discuss this discrepancy in the Discussion section.
  3. What is the antiviral efficacy of PLGA NP [6BIGOE] in vivo? If the authors cannot perform in vivo experiments, a detailed discussion of limitations and future directions is essential.
  4. Figure 3: Western blot quantifications should include statistical analysis (e.g., densitometry ± SD) rather than representative images alone.

Reviewer 2 Report

Comments and Suggestions for Authors

This manuscript encapsulated 6BIGOE into PLGA nanoparticles and confirmed its anti-influenza activity in vitro. This study suggested that 6BIGOE was a potential candidate as an antiviral agent to treat IAV infections.

  1. What is the administration route of the designed nanoparticles? intravenous injection or inhalation? If the particles are given intravenously, system clearance and accumulation in target cells should be considered. If inhalation, the deposition and clearance in airway are needed to be considered.
  2. The idea of using 6BIGOE in IAV infections treatment is good, but according to the results shown in Fig3, there are no significant difference between free 6BIGOE and 6BIGOE /PLAG nanoparticles. The data didn’t support the title “improved pharmacological 2 profile after encapsulation in PLGA nanoparticles”.
  3. The encapsulation efficiency and drug loading capacity are too low, which will affect subsequent in vivo studies. The preparation process needs further optimization.
  4. The release profiles in vitro are needed.

Reviewer 3 Report

Comments and Suggestions for Authors

In this brief report, Schroeder et al, addressed the “Anti-influenza activity of 6BIGOE: improved pharmacological profile after encapsulation in PLGA nanoparticles”. Having examined the manuscript, I note that though it discusses interesting observations, to be considered for MDPI International Journal of Molecular Sciences, the following are some of the comments that the authors might find useful for future submission. This brief report is well-structured in vitro study demonstrating the anti-influenza potential of PLGA-encapsulated 6BIGOE nanoparticles via GSK-3β inhibition. The authors carried out a comprehensive study, including nanoparticle characterization,  cytotoxicity assays (MTT and LDH), and H1N1 replication assays in A549 and Calu-3 cells.  They also confirmed the GSK-3β inhibitory role of PLGA encapsulated 6BIGOE nanoparticles.  This type of studies are extremely  valuable for the scientific community at a global level.

Reviewer Comments

  1. The in vitro findings of PLGA-encapsulated 6BIGOE nanoparticles are promising. Can the authors briefly comment on how these results might translate to more complex in vivo systems?
  2. Can authors provide the information regarding the stability of 6BIGOE in PLGA nanoparticles?
  3. What is the release kinetics of 6BIGOE in PLGA nanoparticles?
  4. Have the authors considered testing the formulation against other influenza virus subtypes (H1N2 and H3N2) or strains to evaluate its broader spectrum of activity? If not, I suggest authors to include this data, as it would further strengthens the manuscript
  5. Do the authors anticipate that this nanoparticle formulation could have antiviral effects beyond H1N1? If so, are there any preliminary data or literature supporting this?
